# Emergence of Plasmid-Mediated Quinolone Resistance (PMQR) Genes in *Campylobacter coli* in Tunisia and Detection of New Sequence Type ST13450

**DOI:** 10.3390/antibiotics13060527

**Published:** 2024-06-05

**Authors:** Manel Gharbi, Rihab Tiss, Melek Chaouch, Safa Hamrouni, Abderrazak Maaroufi

**Affiliations:** 1Group of Bacteriology and Biotechnology Development, Laboratory of Epidemiology and Veterinary Microbiology, Institut Pasteur de Tunis, University of Tunis El Manar (UTM), Tunis 1002, Tunisia; rihab.ettiss95@gmail.com (R.T.); safa.hamrouni@pasteur.tn (S.H.); abderrazak.maaroufi@pasteur.tn (A.M.); 2Laboratory of Medical Parasitology, Biotechnology and Biomolecules (LR16IPT06), Pasteur Institute of Tunis, Tunis 1002, Tunisia; melek.chaouch@pasteur.utm.tn; 3Laboratory of BioInformatics, BioMathematics and BioStatistics (LR16IPT09), Pasteur Institute of Tunis, Tunis 1002, Tunisia

**Keywords:** *Campylobacter coli*, poultry, plasmid mediated quinolone resistance, ST13450

## Abstract

The aim of this study is to investigate the occurrence of plasmid mediated quinolone resistance (PMQR) determinants in *Campylobacter coli* isolates collected from broilers, laying hens and poultry farm environments. One hundred and thirty-nine *C. coli* isolates were isolated from broilers (n = 41), laying hens (n = 53), eggs (n = 4) and the environment (n = 41) of 23 poultry farms located in northeastern of Tunisia. Antimicrobial susceptibility testing was performed on all isolates according to the recommendation of the European Committee on Antimicrobial Susceptibility Testing guidelines. The detection of PMQR genes: *qnrA*, *qnrB*, *qnrC*, *qnrD*, *qnrS*, *qepA*, and *aac*(6)*-Ib* gene was performed using polymerase chain reaction (PCR) and specific primers. *aac*(6′)-Ib amplicons were further analyzed by digestion with *BtsCI* to identify the *aac*(6′)-*Ib-cr* variant. Mutations in GyrA and the occurrence of RE-CmeABC efflux pump were determined by mismatch amplification mutation assay (MAMA) PCR and PCR, respectively. In addition, eleven isolates were selected to determine their clonal lineage by MLST. The 139 *C. coli* isolates were resistant to ciprofloxacin, and 86 (61.8%) were resistant to nalidixic acid. High rates of resistance were also observed toward erythromycin (100%), azithromycin (96.4%), tetracycline (100%), chloramphenicol (98.56%), ampicillin (66.1%), amoxicillin-clavulanic acid (55.39%), and kanamycin (57.55%). However, moderate resistance rates were observed for gentamicin (9.35%) and streptomycin (22.3%). All quinolone-resistant isolates harbored the Thr-86-Ile amino acid substitution in GyrA, and the RE-CmeABC efflux pump was detected in 40.28% of isolates. Interestingly, the *qnrB*, *qnrS*, *qepA*, and *aac*(6′)-*Ib*-*cr* were detected in 57.7%, 61.15%, 21.58%, and 10% of isolates, respectively. The eleven isolates studied by MLST belonged to a new sequence type ST13450. This study described for the first time the occurrence of PMQR genes in *C. coli* isolates in Tunisia and globally.

## 1. Introduction

The *Campylobacter* genus belongs to the phylum *Proteobacteria*, the class *Epsilonproteobacteria*, the order *Campylobacterales*, and the family *Campylobacteraceae*. *Campylobacter* species are Gram-negative; spiral, rod-shaped, or curved; motile; and non-spore-forming. Cells are approximately 0.2–0.8 μm wide and 0.5–5 μm long [1]. The genus includes at least 61 species and 10 subspecies [2]; however, among the 13 pathogenic *Campylobacter* spp. known to be related to human infections, *Campylobacter jejuni* and *Campylobacter coli* are the top two species that are responsible for >95% of infections worldwide and thus are major sources of concern for the general public around the world [3]. Importantly, the majority of human infections are caused by C. jejuni (80–85%), whereas most of the remaining cases are attributed to C. coli [3]. Campylobacteriosis is the name for the contagious illnesses caused by pathogenic *Campylobacter* species, and *Campylobacter* infections are on the rise at a rate greater than shigellosis throughout the world, especially in developing countries [4]. 

Many domestic animals and birds, including commercial broiler chickens, which are mainly considered asymptomatic carriers, harbor *Campylobacter* spp. in their gastrointestinal tracts as part of the normal microbiota [5]. Poultry serves as the natural host for thermophilic *Campylobacter* species, which are the cause of 80% of cases of human campylobacteriosis. In addition, *Campylobacter* contamination in cattle, pigs, pets, wild birds, and other animals has been reported [6]. 

Since *Campylobacter* is a commensal of numerous animal species, it is exposed to a wide variety of classes of veterinary medicines such as quinolones (such as enrofloxacin), macrolides (erythromycin), and tetracyclines. Consequently, as a result of this practice, high rates of resistance have been reported in *C. coli* isolates as well as *C. jejuni* obtained from farms and food chain products. Besides their intrinsic resistance to sulfamethoxazole, trimethoprim, vancomycin, cloxacillin, and oxacillin, several acquired antimicrobial resistances have been reported in *Campylobacter* owing to their natural competence and hypervariable genomic sequences, conferring an important genomic plasticity [7]. 

In *C. coli* as well as *C. jejuni*, multiple mechanisms for antimicrobial resistance have been reported, including (i) production of antibiotic-inactivating/modifying enzymes (e.g., β-lactamase), (ii) alteration or protection of antibiotic targets (e.g., ribosomal protection protein Tet(O), mutations in GyrA or 23S rRNA genes), (iii) active efflux of antimicrobials (e.g., *Campylobacter* multidrug efflux pump), and (iv) reduced permeability to antimicrobials due to unique membrane structures. Over the last decade, several studies have reported very high resistance levels (80–100%) to ciprofloxacin in human *Campylobacter* isolates [8]. Since *Campylobacter* infections are mainly treated with fluoroquinolones and macrolides, fluoroquinolone-resistant *Campylobacter* was listed as one of the six high-priority antimicrobial-resistant pathogens by the World Health Organization in 2017 [9]. From the molecular point of view, resistance to fluoroquinolones is mainly due to amino acid substitution(s) in the quinolone resistance-determining region (QRDR) of the corresponding topoisomerase (DNA gyrase). Several punctual mutations in GyrA associated with fluoroquinolone resistance in *Campylobacter *species include Thr86Ile, Asp90Asn, Thr86Lys, Thr86Ala, Thr86Val, and Asp90Tyr, of which Thr86Ile substitution (C257T in *gyrA* gene) is the most frequently observed mutation and confers a high-level resistance to fluoroquinolones [10]. Fluoroquinolone resistance is also conferred by the CmeABC multidrug efflux pump and the resistant-enhancing variant of the cmeABC efflux pump (RE-CmeABC), which have been described as the major efflux mechanism, causing resistance to several antimicrobial agents including fluoroquinolones and macrolides. Both pumps are the most common efflux systems in *Campylobacter* and work in synergy with GyrA mutations in causing fluoroquinolone resistance [11]. In *Enterobacteriaceae*, besides mutations in QRDR of GyrA and ParC, plasmid-mediated quinolone resistance (PMQR) determinants have been increasingly reported during the last two decades. These PMQR determinants encode several mechanisms, including (i) ribosome protection by QNR proteins, (ii) antibiotic modification by the aminoglycoside modifying enzyme AAC(6′)-Ib-*cr*, and (iii) antibiotic efflux, which is mediated by 2 main transferable efflux pumps (QepA and OqxAB) [12]. Globally, including Tunisia, PMQR determinants have been reported in *Enterbacteriaceae* strains, mainly in *Escherichia coli*, from human and animal origins [13,14]. Since *E. coli* is a normal colonizer of the gastrointestinal tract of several *Campylobacter*’s hosts such as poultry, the transfer of these PMQR from *E. coli* to *Campylobacter* strains is likelihood. 

In Tunisia, several studies have reported high rates of fluoroquinolone resistance in *Campylobacter* strains from poultry origins and chromosomal mutations in GyrA as well as the occurrence of CmeABC have been reported in characterized strains [15,16,17,18,19,20]. Therefore, in this study, we investigated the occurrence of PMQR determinants in previously characterized *C. coli* isolates collected from broilers, laying hens and poultry farm environments.

## 2. Results

### 2.1. Antimicrobial Susceptibility of C. coli Isolates

Among the 139 *C. coli* isolates, all were resistant to ciprofloxacin and 86 (61.8%) were resistant to nalidixic acid (Figure 1). High rates of resistance were also observed toward erythromycin (n = 139, 100%), azithromycin (n = 134, 96.4%), tetracycline (n = 139, 100%), chloramphenicol (n = 137, 98.56%), ampicillin (n = 92, 66.1%), amoxicillin-clavulanic acid (n = 77, 55.39%), and kanamycin (n = 80, 57.55%). However, moderate resistance rates were observed for gentamicin (n = 13, 9.35%) and streptomycin (n = 31, 22.3%).

### 2.2. Screening for PMQR Genes 

All quinolone-resistant isolates harbored the Thr-86-Ile amino acid substitution in GyrA as determined by MAMA PCR. In addition the RE-CmeABC efflux pump was detected in 40.28% (n = 56) of isolates (Table 1). Interestingly, the *qnrB*, *qnrS*, *qepA*, and *aac*(6′)-Ib-*cr* were detected in 57.7%, 61.15%, 21.58%, and 10% of isolates, respectively (Table 1). Interestingly, 14 isolates harbored all the detected six genes (*qnrB*, *qnrS*, *qepA*, *aac*(6′)-Ib-*cr*, and RE-cmeABC). 

### 2.3. MLST Typing

Eleven isolates were selected according to the occurrence of the majority of genes and their antimicrobial resistance profiles, and MLST was performed to determine their genetic relatedness. The eleven isolates belonged to the same sequence type ST13450, which is a new ST. As shown in Figure 2A,B, the ST13450 clustered with the nearest STs corresponding to strains from Tanzania, Peru, Brazil, and Australia. Interestingly, ST13450 did not belong to the internationally known complexes ST283, ST661, ST1150, or ST1332 but to the ST828-complex, the nearest complex (Figure 3). As shown in Figure 4, our isolates were firstly closely linked to isolates derived from human samples (human stool) and then to strains isolated from chickens.

## 3. Discussion

The rise of resistance in infectious pathogens toward clinically relevant antimicrobials is a global threat to human and animal health. Acquired antimicrobial resistances are mainly carried by mobile genetic elements such as plasmids and transposons, as well as complex genetic structures such as integrons, which promote their large spread within bacterial isolates belonging to the same genus or between different genera and even between different families [21]. *Campylobacter* is exposed to antimicrobials used in food-producing animals, companion animals, and humans; thus they have a strong capacity for adaptation to antibiotic selection pressure and have evolved a variety of antibiotic resistance mechanisms [12,22]. *E. coli* is a well-known commensal species of humans and various animals and widely considered a sentinel of antimicrobial resistance phenotypes and genotypes owing to their genome plasticity [23]. This bacterium shares the intestinal tract of *Campylobacter’s* hosts such as poultry, the premium host of thermotolerant *Campylobacter* species. 

The antimicrobial susceptibility of our isolates was performed by the classical disc diffusion method (standard Kirby–Bauer disk diffusion) according to Clinical and Laboratory Standards Institute (CLSI) guidelines. However, other studies investigated the antimicrobial minimal inhibitory concentrations using the broth microdilution method which seems more convenient for comparing resistance levels among the different resistance mechanisms.

To the best of our knowledge, this is the first report describing the occurrence of PMQR determinant in *C. coli* and in *Campylobacter* spp. in general. Indeed, the *qnrB*, *qnrS*, *qepA*, and *aac*(6′)-Ib-*cr* genes were detected relatively at high and unexpected rates, being 57.7%, 61.15%, 21.58%, and 10% of isolates, respectively. The *qnr* genes (*qnrA*, *qnrB*, *qnrC*, *qnrD*, *qnrS*, and *qnrVC*) (https://www.ncbi.nlm.nih.gov/pathogens/refgene/#, accessed on 1 May 2024) code QNR proteins, belonging to the pentapeptide repeat family that bind to DNA gyrase or topoisomerase IV inhibiting the gyrase-DNA interaction [24]. The *qnr* determinants are associated with low-level resistance to quinolones; however, studies have indicated that their presence enhances the selection of chromosomal mutations causing high-level quinolone resistance [24]. Globally and in Tunisia, in *Enterobacteriaceae*, mainly *E. coli*, the *qnrA* and *qnrS* genes have been highly reported from various origins [23,24,25,26]. In a previous study, published in 2005, the *qnrA* genes were absent among 145 *C. jejuni* clinical isolates from Greece [27]. In addition, in that study, the reference strain *E. coli* J53 harboring *qnrA*-carrying plasmid pMG252 was unable to transfer this plasmid either through transformation or through conjugation to the reference strains *C. jejuni* NC 012561 and NC 012539 [27]. It has been proposed that *E. coli* plasmids, including broad host range plasmids, do not replicate in *Campylobacter*, which is most efficiently transformed by plasmid DNA from its own species [28]. However, the *ermB* gene encoding macrolides-lincosamides-streptogramine resistance frequently reported in *Campylobacter* strains is also common and firstly reported in Gram-positive bacteria such as staphylococci and enterococci, indicating possible acquisition of *ermB*-borne plasmids from those Gram-positive bacteria. This might indicates that horizontal transfer of plasmid containing resistance genes probably occur in general and cannot be excluded. 

PMQR had been reported and disseminated in *Enterobacteriaceae* by the beginning of 2000s. The majority of studies performed on *Enterobacteriaceae* species such as *E. coli* and *K. pneumoniae* from various origins mainly focus on the detection of the PMQR genes by PCR without isolation or genetic characterization of the vector plasmids. However, from an epidemiologic point of view, genetic characterization of plasmids carrying PMQR genes is very interesting. This can be determined by hybridization with PMQR gene-probe on agarose gel containing plasmid extract or S1-PFGE (especially for big plasmid (megaplasmid 400–500 kb)). Other probes corresponding to other antimicrobial resistance genes can also be used to investigate their possible genetic linkage on the corresponding plasmids. Unfortunately, southern blot and hybridization technology is not available for all investigators owing to its high cost. 

The plasmidic efflux pump QepA (ten alleles: QepA1-QepA10) confers significantly decreased susceptibility to norfloxacin, ciprofloxacin, and enrofloxacin with 8-fold to 32-fold increases in MIC of each of them [24]. They have been commonly described in *Enterobacteriaceae* isolates, as well as in *Moraxellaceae* and *Pseudomonadaceae* [14]. Therefore, herein this is the first report of its occurrence in *Campylobacter* species.

The aminoglycoside acetyltransferase AAC (6′)-Ib enzyme confers resistance to tobramycin, kanamycin, and amikacin. The *aac*(6′)*-Ib-cr* variant of *aac*(6′)*-Ib* contributes reduced susceptibility to ciprofloxacin by *N*-acetylation of its piperazinyl amine. Therefore it is a bifunctional acetyltransferase, capable of modifying both aminoglycosides and quinolones [29]. Globally, this enzyme is widely reported from Gram-negative bacteria especially in *Enterobacteriaceae* of human and farmed animal [30].

Taken together, the occurrence of various PMQR genes and CmeABC efflux pump in association with punctual mutation in GyrA (Thr-86-Ile), clearly showed the ability of *C. coli* isolates to acquire various molecular mechanisms leading to quinolone/fluoroquinolone resistance as well as resistance to other antimicrobials: (i) *aac*(6′)*-Ib-cr* encoding resistance to both fluoroquinolones and aminoglycosides, and (ii) CmeABC encodes resistance to quinolone/fluoroquinolone and tetracyclines.

As *C. jejuni* mainly co-exist with *C. coli* in gastrointestinal tract of poultry, the emergence of PMQR genes in avian *C. jejuni* isolates is likelihood in the near future not only in the Tunisia but also in other part of the world. It is worthy to note that the current study will encourage researchers worldwide to investigate these PMQR determinants in *Campylobacter* isolates either from human or livestock origins and surely more data will be available concerning the molecular epidemiology of these determinants in the *Campylobacter’s* genetic background.

The MLST methodology is highly reproducible, and the results can be made available and shared in specific databases that make it possible to compare the genetic diversity of bacteria isolated from different countries [31,32]. MLST has been successfully used in *Campylobacter* strain characterization, especially *C. jejuni* species [33]. However, it is important to note that globally, there is a relative paucity of genomic studies on *C. coli* and that there are no published studies using MLST methodology comparing *C. coli* strains circulating in Tunisia among them and with other *C. coli* strains isolated in other countries [15,16,17,18,19].

The eleven studied strains belonged to the same ST13450, which is a new ST. Despite the reduced number of analyzed strains, this finding highlights the genetic homogeneity of *C. coli* strains circulating in avian Tunisian farms. Comparing our isolates to those deposed in PubMLST database showed their high genetic similarity to strains of human and chicken sources, confirming the role of poultry as reservoir and vector of *C. coli* to human. The nearest STs corresponding to strains from Tanzania, Peru, Brazil, and Australia. In addition, this ST does not belong to any of the four internationally known complexes ST283, ST661, ST1150, and ST1332 [31]; however, it was very close to the pandemic ST828-complex [34]. Despite the small number of isolates studied by MLST, it seems that this clone is dominant among quinolone resistant *C. coli* isolates collected from various farms in the north of Tunisia. However, further molecular studies of higher number of isolates are needed to better understand the clonal expansion of circulating quinolone resistant *C. coli* isolates in avian Tunisian’s farms. 

## 4. Materials and Methods

### 4.1. Ethics Statement

The study was approved by the Biomedical Ethics Committee of the Pasteur Institute of Tunis, with reference number: 2018/12/I/LR16IPT03. 

### 4.2. Bacterial Strains

One hundred and thirty nine *Campylobacter coli* isolates were previously taken from broilers (n = 41), laying hens (n = 53), eggs (n = 4) and the environment (n = 41) [17,18,19]. These isolates were recovered from 23 poultry farms located in northeastern of Tunisia between December 2016 and May 2018. Samples were inoculated into Bolton Broth (Oxoid Ltd., Baskingstoke, Hampshire, UK) and incubated, for enrichment, at 42 °C for 48 h in a microaerobic environment (5% O_2_, 10% CO_2_ and 85% N_2_), created by GENbox generators (BioMerieux, Marcy l’Etoile, France). A 10 µL quantity of each enriched sample was streaked on Karmali agar plates (Sigma-Aldrich, Bangalore, India) and incubated at 42 °C for 48 h in a microaerobic environment. From each sample, a total of ten swarming, opaque, white to grey colonies suspected of being *Campylobacter* were subcultured on Karmali agar. Colonies comprising curved or spiral motile rods, when observed by light microscopy, were examined for oxidase/catalase activities and Gram stained. Thereafter, a maximum of three microscopically confirmed *Campylobacter* isolates per sample were subjected to PCR analyses for genus confirmation and species identification [17].

### 4.3. Antimicrobial Susceptibility Testing

Antimicrobial susceptibility testing was performed on all isolates using the disk diffusion method on Mueller-Hinton medium (Bio Life, Milan, Italy) according to the recommendation of the European Committee on Antimicrobial Susceptibility Testing guidelines [35]. The following antimicrobials were used (Oxoid, Basingstocken, UK): ampicillin (10 μg), amoxicillin/clavulanic acid (10/20 μg), tetracycline (30 μg), erythromycin (15 μg), azithromycin (15 μg), gentamicin (10 μg), streptomycin (10 μg), kanamycin (30 μg), chloramphenicol (30 μg), nalidixic acid (30 μg), and ciprofloxacin (5 μg). The isolates were defined as multidrug-resistant (MDR) if they exhibited resistance to at least one agent belonging to three or more antimicrobial families [36].

### 4.4. Screening of PMQR Genes from Campylobacter Isolates

The genomic DNA of collected isolates was extracted using the boiling method [16]. The detection of PMQR genes: *qnrA*, *qnrB*, *qnrC*, *qnrD* and *qnrS* was performed using polymerase chain reaction (PCR) and specific primers [35]. PCR amplifications were performed in a thermocycler (Applied Biosystems, Waltham, MA, USA) as follows: 95 °C for 5 min and 35 cycles of 1 min at 95 °C, 1 min at specific annealing temperature for each primer, and 1 min at 72 °C, and a final extension step of 10 min at 72 °C. Amplification reactions were prepared in a total volume of 25 μL (24 μL of PCR master mix plus 1 μL of template DNA) including 5 ng of genomic DNA, 2.0 U of Taq DNA polymerase (Fermentas, Vilnius, Lithuania), 0.2 mM deoxyribose nucleoside triphosphate mix, 1.50 mM MgCl_2_, 1μM of each primer, and 1X PCR buffer. The *aac*(6)*-Ib* gene was amplified by PCR with primers 5-TTGCGATGCTCTATGAGTGGCTA and 5-CTCGAATGCCTGGCGTGTTT to produce a 482-bp product [36]. PCR conditions were 34 cycles of 94 °C for 45 s, 55 °C for 45 s, and 72 °C for 45 s. Strains positive and negative for *aac*(6)*-Ib* were included as controls. All PCR products positive for *aac*(6′)-Ib were further analyzed by digestion with *BtsCI* (New England Biolabs, Beverly, MA, USA) [37]. PCR products were electrophoresed on 1% agarose gel at 100 V, stained with ethidium bromide solution, and finally visualized in gel documentation system (UVItec Limited, Cambridge, UK). 

### 4.5. Phylogenetic Analysis by Multilocus Sequence Typing (MLST) 

Eleven isolates were selected according to the occurrence of the majority of genes and their antimicrobial resistance profiles to determine their clonal lineage by MLST. PCR amplicons identifying seven allele loci (*aspA*, *glnA*, *gltA*, *glyA*, *pgm*, *tkt*, and *uncA*) were obtained for each isolate by using the primers provided in PupMLST database (https://pubmlst.org/organisms/campylobacter-jejunicoli/primers, accessed on 1 May 2024) [25]. Each 50-μL PCR mixture contained 39.75 μL of molecular biology-grade water (Sigma Aldrich Company Ltd.), 5 μL of 10× PCR buffer (Qiagen, Germany.), 1 μL of 10 μM of each forward and reverse primers, 1 μL of a 10 mM deoxynucleoside triphosphate mixture (Invitrogen Ltd.), 0.25 μL of HotStar Taq DNA polymerase (Qiagen Ltd.), and 2 μL (approximately 10 ng) of *C. coli* chromosomal DNA. The amplification conditions were 95 °C for 15 min, followed by 35 cycles of 94 °C for 30 s, 50 °C for 30 s, and 72 °C for 1 min, with a final extension at 72 °C for 5 min and storage at 4 °C. Nucleotide sequencing was performed with the same primers (diluted 1:15 in water) and 30 cycles of 96 °C for 10 s, 50 °C for 5 s, and 60 °C for 2 min. After sequencing of PCR products ST profiles were assigned by submitting the sequences to the PubMLST database using the submission database. Novel alleles were submitted to the PubMLST *C. jejuni*/C. coli databases curators for number assignment (http://pubmlst.org/campylobacter/, accessed on 1 May 2024). 

The MLST profile of the strains studied was obtained using the online software https://pubmlst.org/bigsdb?db=pubmlst_campylobacter_isolates&page=plugin&name=GrapeTree&l=1, accessed on 1 May 2024, which identifies the allelic sequence of each of the seven housekeeping genes comprising the *Campylobacter* MLST schema in submitted assemblies and compares them with sequences deposited in the PubMLST database (http://pubmlst.org, accessed on 1 May 2024). Thus, allele numbers and allelic profiles were obtained, which were then used to identify the ST (Sequence Type) and CC (Clonal Complex) of each strain studied. 

The result of this analysis was subjected to clustering using the program GrapeTree (http://pubmlst.org/campylobacter/, accessed on 1 May 2024). The resulting phylogenetic tree was subsequently visualized using the Interactive Tree of Life (iTOL) v5 program (available at https://itol.embl.de, accessed on 1 May 2024) annotated to indicate the number of allele differences.

## 5. Conclusions

Taken together, this study describe for the first time the occurrence of PMQR genes in *C. coli* isolates in Tunisia and in the world. This finding highlights the ability of *Campylobacter* to acquire various molecular mechanisms not only against quinolone/fluoroquinolones but also against other antimicrobial agents.

## Figures and Tables

**Figure 1 antibiotics-13-00527-f001:**
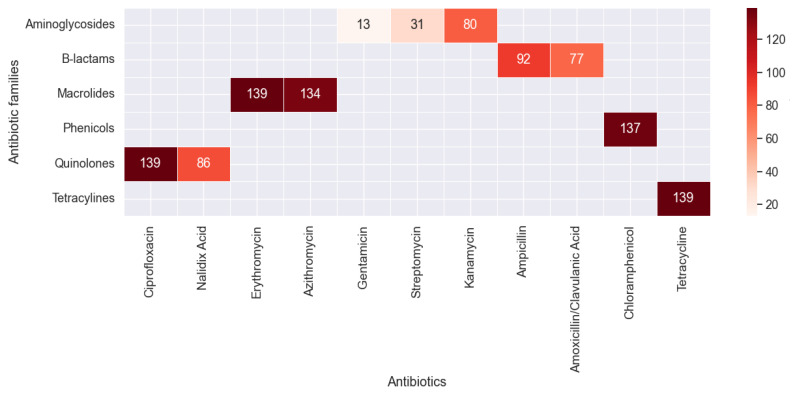
Rates of resistance toward tested antimicrobials in the 139 *C. coli* isolates.

**Figure 2 antibiotics-13-00527-f002:**
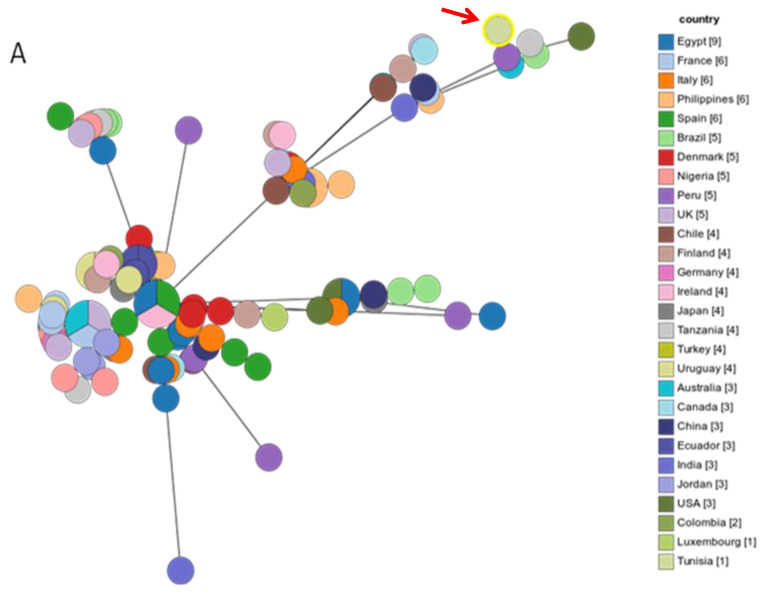
Genetic relationships of the *C. coli* ST13450 lineage detected in this study and *C. coli* isolates from the world (different STs from each country) in the PubMLST database (data taken in April, 2024). (**A**): a minimum spanning tree was reconstructed based on the ST from this study and the MLST database. The size of circles is proportional to the number of isolates of the same ST in different countries, and the sources of the isolates are colored as indicated. (**B**): Distribution of STs according to the countries, the size of nodes indicates the number of isolates with identical ST in the country.

**Figure 3 antibiotics-13-00527-f003:**
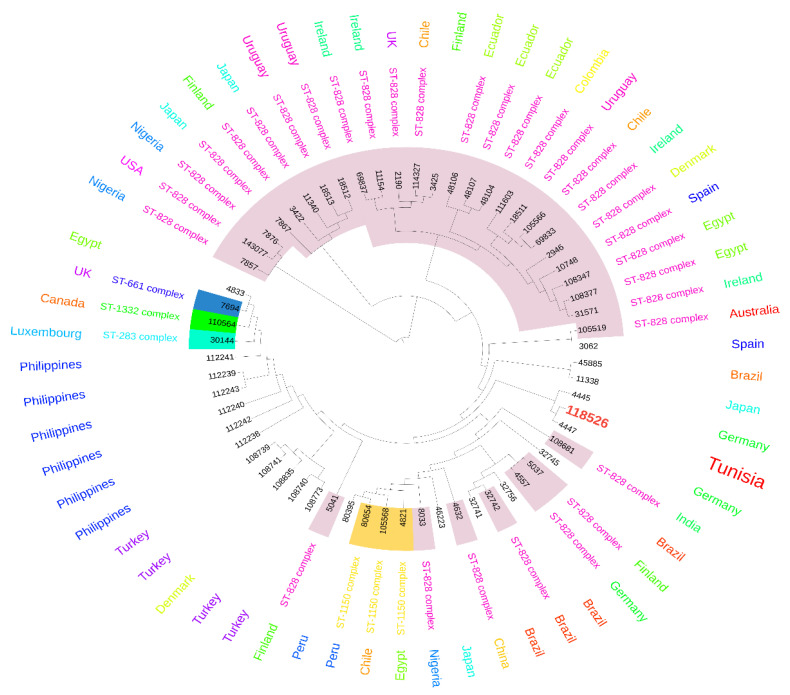
Phylogenetic tree based on MLST sequences and allelic profiles of *C. coli*. Isolates of undetermined STs with allelic profiles deposed in PubMLST database were also included.

**Figure 4 antibiotics-13-00527-f004:**
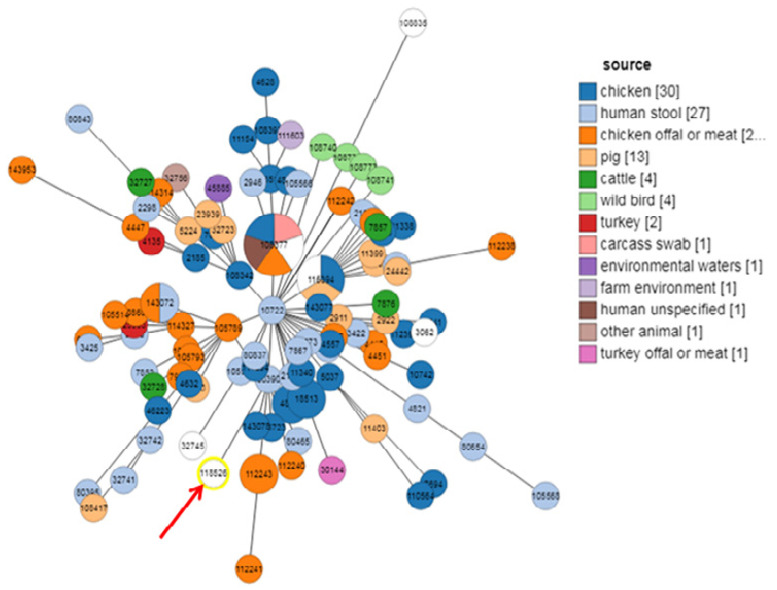
Minimum Spanning Tree of the ST data from *C. coli* sourced from PubMLST (Different STs from each country). Circle diameter is proportional to the number of isolates of the same ST in different sample sources in the PubMLST database and color depicts sources as indicated in the legend.

**Table 1 antibiotics-13-00527-t001:** Occurrence of PMQR and CmeABC genes in the 139 *C. coli* isolates.

Origins of Isolates	*qnrA* n (%)	*qnrB* n (%)	*qnrC* n (%)	*qnrD* n (%)	*qnrS* n (%)	*qepA* n (%)	*aac*(6′)-*Ib* n (%)	*aac*(6′)-*Ib*-*cr* n (%)	RE-*cmeABC* n (%)
Broiler (n = 41)	0	23 (56)	0	0	23 (56)	0	31 (75.6)	10 (24.39)	25 (60)
Layer (n = 53)	0	35 (66)	0	0	33 (62.3)	17 (32)	11 (26.86)	4 (9.7)	19 (35.84)
Eggs (n = 4)	0	4 (100)	0	0	2 (50)	1 (25)	0 (0)	0 (0)	3 (75)
Environment (n = 41)	0	18 (43.9)	0	0	27 (65.8)	12 (29.26)	0 (0)	0 (0)	9 (21.95)
Total (n = 139)	0	80 (57.7)	0	0	85 (61.15)	30 (21.58)	42 (30.21)	14 (10)	56 (40.28)

## Data Availability

The statistical data used to support the findings of this study are available from the corresponding author upon request.

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
