# Peer review of "Emergence of Plasmid-Mediated Quinolone Resistance (PMQR) Genes in Campylobacter coli in Tunisia and Detection of New Sequence Type ST13450"

_antibiotics, 2024, doi:10.3390/antibiotics13060527_

Round 1

Reviewer 1 Report

Comments and Suggestions for Authors

It is an interesting study of antibiotic resistance for C. coli. 

It should be updated with some scientific works from the years 2023 and 2024 on Pub Med. Conclusions should be presented either separately or inserted at the end of the "Discussions" section.

Author Response

Reviewer 1

It should be updated with some scientific works from the years 2023 and 2024 on Pub Med. Conclusions should be presented either separately or inserted at the end of the "Discussions" section.

RESPONSE: as you suggested we added a conclusions section at the end of the manuscripts (4. Conclusions: Taken together, this study describe for the first time the occurrence of PMQR genes in C. coli isolates in Tunisia and in the world. This finding highlights the ability of Campylobacter to acquire various molecular mechanisms not only against quinolone/fluoroquinolones but also against other antimicrobial agents).

All recent studies (2023-2024) did not report the occurrence of PMQR in Campylobacter, that is why we said that this is the first report of PMQR genes in campylobacter.

Reviewer 2 Report

Comments and Suggestions for Authors

This ms reports plasmid-mediated quinolone resistance (PMQR) genes in Campylobacter coli of poultry sources/environment in Tunisia. The applied approach was straightforward with the susceptibility testing method less useful. The findings are useful for understanding of fluroquinolone resistance in C. coli but the limitations of the study should be discussed. Comments are included below.

The ms focuses on PMQR C. coli in poultry. Yet, plasmids were not obtained. Were there some so-called plasmidborne genes in the chromosomes? Please discuss.

Additional information on C. jejuni is also needed (Introduction/Discussion).

L100/L234. Disk method, not the broth dilution method, was used for antibiotic susceptibility testing. This obviously is weak for comparing resistance levels among the different resistance mechanisms. This also explains the problem for 100% ciprofloxacin resistance versus only 62% nalidixic acid resistance (L101). This ms should comment on the method for its limitation. Please remove all abbreviations used such as “AMP”, “AMC”, “TET”, “k”.. “CIP” (L239-242) as they are shown elsewhere in the ms.

L109. Please include and describe how many isolates are detected with more than one resistance gene,

L116. Why only 11 isolates were selected? The limitations should be addressed.

L146. Would also include poultry C. jejuni and C. coli distribution situation. More critically, this reviewer, like other readers, would be more interested in what would happen to C. jejuni from the same region/time with respect to PMQR. This point should be discussed to improve the ms. As well, the disk method had made it impossible to compare resistance levels mediated by different genes, a major deficiency of this investigation. A key opportunity for a much better understanding of C. coli resistance is lost. Even this ms tried to compare MIC fold changes using literature data, it is inadequate. Thus, one cannot know what major mechanisms play major role in quinolone resistance in C. coli (which is known to have high intrinsic resistance to multiple classes of antimicrobials).

229. Please include the year information for the isolates tested, and more importantly how these strains were identified, even the references have the information. The strain identification is critical. How these isolates were selected for PMQR study as all isolates were resistant to ciprofloxacin. The readers would also be interested the quinolone resistance in C. jujuni and C. coli of poultry. Please include additional information to improve the ms.

All figures (1 to 4). Very poor resolution. Must be changed.

Minors (examples)

L20. Not sure why “qepA” (seen in L30) is not mentioned.

L23.  Spell out “MAMA” - Mismatch amplification mutation assay (MAMA).

L42-42. No need to have “(C. jejuni” and “(C. coli)”. More importantly, please have some information in appropriate place in Introduction on the C. jejuni as the major species in poultry while C. coli as the major in swine.

L55. “ciprofloxacin” is generally not a veterinary drug. No countries approve its use in animals, at east in food animals.

L65. Would be Tet(O), not “TetO.

L66. Would not use “CME”.

L68. No need to have “a member of fluoroquinolones,”, the drug is induced in L55.

L74. Write “DNA”, not French “ADN”.

L78. Not “encoded” but “conferred”. Please define “RE” first, i.e., resistance-enhancing.

L116. :antimicrobial: is noted, while “antibiotic(s)’ is used in various places. You chose one for consistency in the ms.

L118/122. Write “Figures”, not “figure”; “Figure 4”, not “figure 4”.

L147. Write “antimicrobial resistance” without a dash.

L191. Italicise “N”.

L248. The full spelling of “PCR” would be introduced earlier in the ms when “PCR” was first used or no need to fully spell it.

Comments on the Quality of English Language

see comments above.

Author Response

Reviewer 2

This ms reports plasmid-mediated quinolone resistance (PMQR) genes in Campylobacter coli of poultry sources/environment in Tunisia. The applied approach was straightforward with the susceptibility testing method less useful. The findings are useful for understanding of fluroquinolone resistance in C. coli but the limitations of the study should be discussed. Comments are included below.

  • The ms focuses on PMQR C. coli in poultry. Yet, plasmids were not obtained. Were there some so-called plasmid borne genes in the chromosomes? Please discuss. Additional information on C. jejuni is also needed (Introduction/Discussion

RESPONSE:

* Genes encoding acquired resistance to quinolone/fluoroquinolone are called ‘plasmid mediated quinolone resistance determinants (genes)’ since are mainly carried by Plasmids. The majority of studies on acquired resistance to quinolone/fluoroquinolone in enterobacteria do not perform plasmid extraction; they only perform PCRs to detect those genes. I am agreeing with you, that studying plasmid is also interesting since we can detect the exact plasmids harboring these genes in one isolate. This can be determined by hybridization on plasmid extract or S1-PFGE (for big plasmid (megaplasmid 400 – 500 kb)); however, unfortunately we have not the kit to perform this hybridization method now, but we would like to perform it in the future according to our financial budget. Also I would like to confirm your idea, some studies showed for examples the aac(6’)-Ib-cr can be carried by the chromosome and carried in class 1 integrons. Also some plasmids can co-integrate on the chromosome. According to your comments we added these sentences in the discussion section: PMQR have been reported and disseminated in Enterobacteriaceae by the beginning of 2000s. The majorities of studies performed on Enterobacteriaceae species such as E. coli and K. pneumoniae from various origins mainly focus on the detection of the PMQR genes without isolation or genetic characterization of the vector plasmids. However, in epidemiologic point of view, genetic characterization of plasmids carrying PMQR genes is very interesting. This can be determined by hybridization with PMQR gene-probe on agarose gel containing plasmid extract or S1-PFGE (especially for big plasmid (megaplasmid 400 – 500 kb)). Other probes corresponding to other antimicrobial resistance genes can also be used to investigate their possible genetic linkage on corresponding plasmids. Unfortunately, southern blot and hybridization technology is not available for all investigators owing to its high cost’.

* The study focused on C. coli that is why few data were provided about C. jenuni. In discussion, we added the following sentences concerning C. jejuni. As C. jejuni mainly co-exist with C. coli in gastrointestinal tract of poultry, the emergence of PMQR genes in avian C. jejuni isolates is likelihood in the near future not only in the Tunisia but also in other part of the world. It is worthy to note that the current study will encourage researchers worldwide to investigate these PMQR determinants in Campylobacter isolates either from human or livestock origins and surely more data will be available concerning the molecular epidemiology of these determinants in the Campylobacter’s genetic background.

L100/L234. Disk method, not the broth dilution method, was used for antibiotic susceptibility testing. This obviously is weak for comparing resistance levels among the different resistance mechanisms. This also explains the problem for 100% ciprofloxacin resistance versus only 62% nalidixic acid resistance (L101). This ms should comment on the method for its limitation. Please remove all abbreviations used such as “AMP”, “AMC”, “TET”, “k”.. “CIP” (L239-242) as they are shown elsewhere in the ms.

RESPONSE: You have reason about the broth dilution method (BDM) determining the MICs of tested antibiotics. Yes there are several studies that actually used BDM and others studies use the classical disc diffusion method (standard Kirby–Bauer disk diffusion) according to Clinical and Laboratory Standards Institute (CLSI) guidelines (CLSI . Performance Standards for Antimicrobial Susceptibility Testing. 31st ed. Clinical and Laboratory Standards Institute (M100e); Wayne, PA, USA: 2021. CLSI supplement M100.). To highlight this point, we added this sentence in the discussion section: ‘The antimicrobial susceptibility of our isolates was performed by the classical disc diffusion method (standard Kirby–Bauer disk diffusion) according to Clinical and Laboratory Standards Institute (CLSI) guidelines. However, other studies investigated the antimicrobial minimal inhibitory concentrations using the broth microdilution method which seems more convenient for comparing resistance levels among the different resistance mechanisms.’

Abbreviation of antibiotics was deleted.

L109. Please include and describe how many isolates are detected with more than one resistance gene.

RESPONSE: we added the following sentences concerning the co-occurrence of genes: Interestingly, 14 isolates harbored all the detected six genes (qnrB, qnrS, qepA, aac(6’)-Ib-cr, and RE-cmeABC).

L116. Why only 11 isolates were selected? The limitations should be addressed.

RESPONSE: As you know it is very expensive to perform MLST for all isolates. That is why we studied only 11 isolates. However, these isolates were not randomly selected; we added this sentence in the section materials and methods of MLST: ‘Eleven isolates were selected according to the occurrence of the majority of genes and their antimicrobial resistance profiles’. By this criterion of selection, we believe that they can be representative of the studied population of C. coli.

L146. Would also include poultry C. jejuni and C. coli distribution situation. More critically, this reviewer, like other readers, would be more interested in what would happen to C. jejuni from the same region/time with respect to PMQR. This point should be discussed to improve the ms. As well, the disk method had made it impossible to compare resistance levels mediated by different genes, a major deficiency of this investigation. A key opportunity for a much better understanding of C. coli resistance is lost. Even this ms tried to compare MIC fold changes using literature data, it is inadequate. Thus, one cannot know what major mechanisms play major role in quinolone resistance in C. coli (which is known to have high intrinsic resistance to multiple classes of antimicrobials).

RESPONSE: as mentioned above we added this paragraph about C. jejuni, in the discussion : The study focused on C. coli that is why few data were provided about C. jenuni. In discussion, we added the following sentences concerning C. jejuni. As C. jejuni mainly co-exist with C. coli in gastrointestinal tract of poultry, the emergence of PMQR genes in avian C. jejuni isolates is likelihood in the near future not only in the Tunisia but also in other part of the world. It is worthy to note that the current study will encourage researchers worldwide to investigate these PMQR determinants in Campylobacter isolates either from human or livestock origins and surely more data will be available concerning the molecular epidemiology of these determinants in the Campylobacter’s genetic background.

L229. Please include the year information for the isolates tested, and more importantly how these strains were identified, even the references have the information. The strain identification is critical. How these isolates were selected for PMQR study as all isolates were resistant to ciprofloxacin. The readers would also be interested the quinolone resistance in C. jujuni and C. coli of poultry. Please include additional information to improve the ms.

All figures (1 to 4). Very poor resolution. Must be changed.

RESPONSE: In this study we focused only on C coli isolates and now similar work on C. jejuni. we modified the paragraph as follows: ‘One hundred and thirty nine Campylobacter coli isolates have been previously reported from broilers (n=41), laying hens (n=53), eggs (n=4) and environment (n=41) [17-19]. These isolates were recovered from 23 poultry farms located in northeastern of Tunisia between December 2016 and May 2018. Samples were inoculated into Bolton Broth (Oxoid, UK) and incubated, for enrichment, at 42°C for 48 h in a microaerobic environment (5% O2, 10% CO2 and 85% N2), created by GENbox generators (BioMerieux, France). A 10 µl of each enriched sample was streaked on Karmali agar plates (SIGMA-ALDRICH) and incubated at 42°C for 48 h in a microaerobic environment. From each sample, a total of ten swarming, opaque, white to grey colonies suspected of being Campylobacter were subcultured on Karmali agar. Colonies comprising curved or spiral motile rods, when observed by light microscopy, were examined for oxidase/catalase activities and Gram stained. Thereafter, a maximum of three microscopically confirmed Campylobacter isolates per sample were subjected to PCR analyses for genus confirmation and species identification [17].’

  • I make the figures bigger; I believe readers can understand those figures.

Minors (examples)

L20. Not sure why “qepA” (seen in L30) is not mentioned.

RESPONSE: you have reason; I added this gene in the abstract.

L23.  Spell out “MAMA” - Mismatch amplification mutation assay (MAMA).

RESPONSE: This was corrected in the abstract (..efflux pump were determined by Mismatch amplification mutation assay (MAMA) PCR and PCR, respectively.)

L42-42. No need to have “(C. jejuni” and “(C. coli)”. More importantly, please have some information in appropriate place in Introduction on the C. jejuni as the major species in poultry while C. coli as the major in swine.

RESPONSE: In this sentence I deleted ‘(C. jejuni)” and “(C. coli)”. The sentence was modified as follows: ‘...however, among the 13 pathogenic Campylobacter spp. known to be related to human infections, Campylobacter jejuni and Campylobacter coli are the top two species that are responsible for >95 % of infections worldwide and thus are major sources of concern for the general public around the world [3].  Importantly, the majority of human infections are caused by C. jejuni (80-85%), whereas most of the remaining cases are attributed to C. coli [3].’

L55. “ciprofloxacin” is generally not a veterinary drug. No countries approve its use in animals, at east in food animals.

RESPONSE: in this sentence we provide the name of ciprofloxacin as an example of antibiotic that belong to the family or group of quinolone/fluoroquinolone we do not mean that it is used in veterinary medicine. However, if an antibiotic belonging to quinolone is used in livestock/veterinary medicine if it select resistance to this antibiotic it will also select resistance to other antibiotics from the same family since generally all those antibiotics have the same target (co-resistance). To avoid the point that you mentioned, we delete the term ‘ciprofloxacin’: ‘...classes of veterinary medicines such as quinolones (such as enrofloxacin),......’

L65. Would be Tet(O), not “TetO.

RESPONSE: this was corrected according to your comments.

L66. Would not use “CME”.

RESPONSE: this was deleted.

L68. No need to have “a member of fluoroquinolones,”, the drug is induced in L55.

RESPONSE: this was deleted.

L74. Write “DNA”, not French “ADN”.

RESPONSE: this was corrected (DNA).

L78. Not “encoded” but “conferred”. Please define “RE” first, i.e., resistance-enhancing.

RESPONSE: these were corrected (...Fluoroquinolone resistance is also conferred by the CmeABC multidrug efflux pump and the resistant-enhancing variant of the cmeABC efflux pump  (RE-CmeABC), which have been described as the major efflux mechanism...)

L116. :antimicrobial: is noted, while “antibiotic(s)’ is used in various places. You chose one for consistency in the ms.

RESPONSE: in all the manuscript, I select to use the terms ‘:antimicrobials’ and ‘antimicrobial agents’

L118/122. Write “Figures”, not “figure”; “Figure 4”, not “figure 4”.

RESPONSE: this was corrected

L147. Write “antimicrobial resistance” without a dash.

RESPONSE: this was corrected

L191. Italicise “N”.

RESPONSE: this was corrected.

L248. The full spelling of “PCR” would be introduced earlier in the ms when “PCR” was first used or no need to fully spell it.

RESPONSE: this was corrected in the section ‘Abstract’.

Reviewer 3 Report

Comments and Suggestions for Authors

In this work the Authors investigated the prevalence of plasmid-mediated quinolone resistance (PMQR) genes in 131 C. coli isolates obtained mainly from broilers, laying hens and environment from 23 poultry farms located in Tunisia. Additionally, determined the genetic relatedness of 11 C. coli isolates by MLST typing.

The Authors described for the first time the occurrence of PMQR genes in C. coli in Tunisia and globally. They showed that many of the C. coli isolates possessed PMQR genes, especially qnrB and gnrS. The results of this study provide a novel and essential data to the existing knowledge.

Minor concerns:

- line 217-218 – in this work the MLST type was determined only in 11 isolates (less than 10% of C. coli isolates). and all isolates belonged to the same sequence type ST13450.  Why in your opinion “it seems that this clone is dominant among C. coli isolates collected from farms in Tunisia”??

- line 56-57 – change the sentence as “…high rates of resistance have been reported in C. coli isolates obtained from farms and food chain products.”  

Author Response

In this work the Authors investigated the prevalence of plasmid-mediated quinolone resistance (PMQR) genes in 131 C. coli isolates obtained mainly from broilers, laying hens and environment from 23 poultry farms located in Tunisia. Additionally, determined the genetic relatedness of 11 C. coli isolates by MLST typing.

The Authors described for the first time the occurrence of PMQR genes in C. coli in Tunisia and globally. They showed that many of the C. coli isolates possessed PMQR genes, especially qnrB and gnrS. The results of this study provide a novel and essential data to the existing knowledge.

Minor concerns:

- line 217-218 – in this work the MLST type was determined only in 11 isolates (less than 10% of C. coli isolates). and all isolates belonged to the same sequence type ST13450.  Why in your opinion “it seems that this clone is dominant among C. coli isolates collected from farms in Tunisia”??

RESPONSE: As you know it is very expensive to perform MLST for all isolates. That is why we studied only 11 isolates. However, these isolates were not randomly selected (‘Eleven isolates were selected according to the occurrence of the majority of genes and their antimicrobial resistance profiles’. By this criterion of selection, we believe that they can be representative of the studied population of C. coli. Despite the low number of selected isolates, which were also from different farms, MLST showed that all isolates belong to the same ST (ST13450) that is why we believe that ‘this clone is dominant among C. coli isolates collected from farms in Tunisia’. To avoid any misunderstanding, we specifically add these word ‘quinolone resistant C. coli isolates’ to say that this clone is dominant and quinolone resistant isolates and might be not dominant in quinolone susceptible isolates. The sentence was modified as follows: ‘ Despite, the few number of isolates studied by MLST, it seems that this clone is dominant among quinolone resistant C. coli isolates collected from various farms in the north of Tunisia. However, further molecular studies of higher number of isolates are needed to better understand the clonal expansion of circulating quinolone resistant C. coli isolates in avian Tunisian’s farms’.

- line 56-57 – change the sentence as “…high rates of resistance have been reported in C. coli isolates obtained from farms and food chain products.” 

RESPONSE: this sentence was changed.